# Lactoferrin combined with Coenzyme Q10 ameliorate sarcopenia in an aging mouse model induced by D-galactose

Wenbin Wu[1][☉], Yinhua Zhu[1,2][☉], Yanan Fu[3][☉], Hongfei Xing[4], Xinlu Guo[1], Jichao Xu[5], Wenhui Hu[5], Mingyang Cui[1], Jiaxin Shi[1], Ling Li[5], Weiwei Wang[5], Peng An[1], Yongting Luo[1]*, Junjie Luo[1]*, Qingchang Xing[5]*

1 Department of Nutrition and Health, China Agricultural University, Beijing, China, 2 Food Laboratory of Zhongyuan, Luohe, China, 3 Division of Neonatology, New Century Women's and Children's Hospital, Beijing, China, 4 Beijing University of Chinese Medicine, Chaoyang, China, 5 The Eighth Medical Center of PLA General Hospital, Beijing, China

☉ These authors contributed equally to this work.
* xqc1025@163.com (QX); luojj@cau.edu.cn (JL); luo.yongting@cau.edu.cn (YL)

## Abstract

Sarcopenia is an age-related condition with a slow and prolonged decrease in muscular mass, strength, and function. As the population ages, the frequency of sarcopenia rises, and aggressive prevention methods and effective treatment options are in urgent need. Here, we explore the hypothesis that nutritional interventions can ameliorate skeletal muscle aging in mice affected by sarcopenia, and the aforementioned hypothesis was validated through histopathological characterization and behavioral experiments. The model group exhibited reduced muscle mass (Lean Mass, GAS Index), muscular strength (Maximum Limb Muscle Strength), and muscular function (Exhaustion Time, Inverted Grid Time), along with increased fat content and smaller myofiber size compared to the control group. Treatments with lactoferrin and CoQ10, both individually and in combination, enhanced muscle indices and facilitated muscle tissue regeneration, with the combined treatment showing the most significant improvement. Research further shows that Lactoferrin and CoQ10, whether administered alone or in combination, were discovered to restrain the progression of sarcopenia by inhibiting both protein metabolism and mitochondrial energy metabolism, and compared to groups treated with lactoferrin or CoQ10 alone, the combined treatment demonstrated varying degrees of improvement across all evaluated metrics, such as Lean Mass (2.273~5.365%), Fat Mass (−1.058~−0.359%), GAS index (0.259~0.335%), Maximum Limb Muscle Strength (6.83~53.498 g), Inverted Grid Time (563~859 s), Exhaustion Time (386~468 s).

**Data availability statement:** All relevant data are within the paper and its Supporting information files.

**Funding:** This work was supported by the National Key Research and Development Program of China (2023YFF1103501).

**Competing interests:** The authors have declared that no competing interests exist.

**Abbreviations:** D-gal, D-galactose; CoQ10, Coenzyme Q10; GAS, Gastrocnemius; TA, Tibialis anterior; EDL, Extensor digitorum longus; SOL, Soleus; HE, Hematoxylin-Eosin; DEGs, Differentially expressed genes; FDR, False discovery rate; GO, Gene Ontology; KEGG, Kyoto encyclopedia of genes and genomes; CAMs, Cell adhesion molecules; EWGSOP, The European Working Group on Sarcopenia in Older People.

## Introduction

In 1989, Dr Irwin Rosenberg coined the term "sarcopenia", from the Greek word for "poverty of flesh" [1] and originally defined it as the age-related loss of muscular mass. Around 2010, sarcopenia was defined as an age-related loss of skeletal muscular mass and strength. Later, muscular strength and physical performance muscular mass were added to the concept of sarcopenia [2]. In the 2018 update of the definition, EWGSOP2 identifies low muscular strength as a key feature of sarcopenia [3]. Sarcopenia is currently defined as a complex degenerative condition characterized by a reduction in skeletal muscular mass, strength, and function that is associated with aging muscular mass, more common in older adults and associated with an elevated likelihood of adverse consequences such as fractures, disability and death [4]. The pathogenesis of sarcopenia is highly complex and is associated with multiple factors such as aging, protein homeostasis, mitochondrial dysfunction [5], dysfunction of satellite cells, inflammation, and insulin resistance [6,7].

The most prevalent interventions for sarcopenia are exercise-based therapy and dietary intervention measures [8]. The effectiveness of kinesitherapy in ameliorating sarcopenia has been extensively demonstrated to lower the incidence of age-related chronic condition, improve quality of lifestyle and even augment the average and maximum life expectancies for human beings [9]. Nutritional intervention is a gentle approach that brings with it scarcely any side effects and holds a vast realm of applications [10], and it can be utilized as an adjunct to physical exercise therapy. Recent research has indicated that nutrients such as BCAAs, HMB and vitamin C exerts a beneficial effect on postponing muscle loss [11–13]. Nevertheless, the current evidence supporting the efficacy of combined or integrated nutritional interventions, remains weak and is in urgent need for additional research in this field [8]. CoQ10 is a fat-soluble compound that occurs naturally in the human body. Human CoQ10 is mainly dependent on its own synthesis and dietary supplementation. CoQ10 is relatively abundant in fish, animal intestines (heart, liver, kidney), beef, pork, peanuts and other foods [14]. CoQ10 can eliminate free radicals in the body [15], relieve physical fatigue, boost immunity, regulate blood lipids and prevent atherosclerosis. Dara M et al. investigated the potentially toxic effects of Sunset Yellow (SY), an artificial food dye, on the testicles of male rats, as well as the improvement of testicular structural abnormalities caused by SY when co-administered with coenzyme Q10. These abnormalities, including spermatogonial degeneration, tubule atrophy, and interstitial tissue damage, further confirm the potential of Coenzyme Q10 as an antioxidant in protecting the male reproductive system from oxidative stress [16]. Meanwhile, mitochondria-targeted therapy drugs have also been demonstrated to be effective in treating diseases associated with cancer and mitochondria [17–23]. In muscular cells, CoQ10 is a crucial component of the mitochondrial electron transport chain, engaged in the production of cellular energy and assists in maintaining the health and function of muscle cells by facilitating the production of ATP (adenosine triphosphate) [24]. Supplementing with protein is also thought to be an important way of helping to slow down muscle loss. Studies have shown that lactoferrin can have a positive effect on muscle cell proliferation, differentiation and resistance to muscle fatigue

[25]. Lactoferrin (Lf) is a multifunctional protein with significant implications for both cardiovascular and bone health. As a source of orally administered antihypertensive peptides, Lf contributes to the regulation of blood pressure, a key factor in maintaining cardiovascular well-being [26]. Beyond its cardiovascular benefits, Lf has been identified as a novel bone growth factor, playing a pivotal role in bone development and regeneration. It facilitates osteoblast proliferation and differentiation while suppressing osteoclast formation, thereby supporting bone anabolic processes [27]. Mechanistically, Lf binds to low-density lipoprotein receptor-related protein 1 (LRP1) on osteoblasts, activating the ERK1/2 pathway to enhance osteoblast proliferation [28]. Additionally, it triggers the p42/44 MAPK signaling pathway in primary osteoblasts, further emphasizing its regulatory influence on bone metabolism [29]. Previous studies have demonstrated that the combination of CoQ10 and lactoferrin (LF) significantly mitigated thioacetamide (TAA)-induced liver and kidney damage in adult male rats. This combined therapy not only effectively improved liver and kidney function and reduced oxidative stress markers (MDA and NOx) levels, but also significantly enhanced the activities of antioxidant enzymes (SOD and CAT). Furthermore, the combination showed a more pronounced effect in alleviating histopathological and ultrastructural damage in the liver and kidneys, and it significantly downregulated the expression of the WNT4 gene associated with kidney injury [30]. However, the involvement of CoQ10 and lactoferrin in the loss of muscles still insufficiently understood.

Decrease in muscular mass, weakening of muscular strength and deterioration of muscle function are the three main characteristics of sarcopenia. Although nutritional intervention plays a crucial role in the prevention and treatment of sarcopenia, it still faces several challenges, including individual variability, limited efficacy of supplements and a lack of comprehensive intervention strategies. However, research has shown that both lactoferrin and CoQ10 significantly influence the process of muscle aging [24,25]. Lactoferrin can improve the supply of protein required for growth of muscular tissue, whilst supplementation of CoQ10 has been proven to boost the supply of ATP, thus yielding substances and fuel for muscle development. Both have the potential to ameliorate sarcopenia in diverse ways. However, it is uncertain if lactoferrin and CoQ10 function alone or in conjunction in sarcopenia. However, the combination of the two can compensate for the lack of protein and energy in people with sarcopenia. The innovation of this study is that for the first time in a mouse model of sarcopenia created by D-gal, the combined intervention of lactoferrin and CoQ10 as a new intervention substance.. The impacts and mechanism of Lactoferrin and CoQ10 on sarcopenia were probed from three aspects: muscular mass, muscular strength, and muscle function. This study not only provides a new perspective for understanding the synergistic effect of these two substances in sarcopenia, but also provides a scientific basis for developing new joint intervention strategies. This combined intervention approach is expected to become a new direction for the treatment of sarcopenia in the future, offering new possibilities for improving muscle health in the elderly and patients with chronic disease.

## Material and methods

### Animals

We selected six-week-old male C57BL/6 mice with a body weight of approximately 20–22 g for the experiment, sourced from SPF Biotechnology Co., Ltd in Beijing, China. The mice were randomly divided into five groups with seven animals in each group.

Throughout the experiment, the mice in each group had unrestricted access to food and water, and their body weight was measured on a weekly basis. The control group was injected with normal saline. For the other groups, D-gal (G0750, Sigma, Germany) was injected at a dosage of 500 mg/kg body weight. The intervention group was administered Lactoferrin (SL9590, Solarbio, China) and CoQ10 (C9538, Sigma, Germany) either alone or in combination by gavage for a duration of 8 weeks [31,32]. With doses of 500 mg/kg/day for Lactoferrin [32] and 10 mg/kg/day for CoQ10 [33], respectively. After eight-week period, the body composition and behavioral characteristics of the mice were measured. Subsequently, the mice were anesthetized by intraperitoneal injection of ketamine (75 mg/kg) and thiazine (8 mg/kg), respectively, and then euthanized with cervical dislocation. Skeletal muscles such as the gastrocnemius (GAS) and tibialis anterior (TA),

were sampled and weighed. Subsequently, relevant histopathological investigations and molecular experiments were carried out. The approval document of this research project comes from the Animal Experiment Ethics Committee of China Agricultural University (AW52404202-5-1). The experiment was conducted in strict compliance with animal ethical principles.

### Analysis of body composition

After an 8-week intervention period, Core NMR And MRI Analyzer MesoMR23-060H-I (QMR23-060H-I, Niumag, China) was employed to measure the body composition of the mice. Each mouse needs to be weighed first and input it into the system before it can be placed in the scanning instrument for measurement. After the measurement is completed, the relevant body composition data of mice can be obtained, including lean meat content, fat content, and free water content, etc.

### Inverted grid test to determine changes in muscular strength

The inverted screen is a wire mesh square with a side length of 43 cm. It is made up of squares with sides of 12 mm and wire with a diameter of 1 mm. It is surrounded by a wooden beading that is 4 cm deep. Each animal was positioned at the grid's center, then inverted 180 degrees and placed 40 cm above the padded cage, and the time the animal remained on the wire before falling was recorded by a timer. Each mouse was tested twice after a 30-minute rest interval. Finally score according to the established criteria.

The scoring criteria set according to the actual situation are as follows: the suspension time (in seconds) divided by 10 yields the score value, out of 100 points. For example, 200 seconds would be recorded as 20 points. Scores above 1000 seconds are also capped at 100 points.

### Grip strength test

The assessment of grip strength was conducted utilizing an electronic grasping force tester (Beijing zhongshidichuang Science and technology development Co., Ltd, China), which provides a precise measurement of the force in the mice's limbs. To measure the grip strength, firmly grasp the midsection of the rat's tail and position it on the elastic metal rod. Ensure that the mouse's limbs establish contact with and securely grasp the elastic metal rod of the force transducer.

The tail of each mouse was gently pulled backwards and make sure to keep the torso and metal bar level, at this point, the tension reading of the electronic grip dynamometer was defined as the grip force of the rat before releasing the elastic metal bar. Each mouse was tested for 3 consecutive times and the maximum limb muscular strength value (g) was calculated.

### Endurance test

Table 1 presents the parameters for the animal experimental treadmill (Beijing zhongshidichuang Science and technology development Co., Ltd, China) during the acclimatization period. The treadmill is configured to a current intensity of 0.3 mA., utilizing a gradual approach to familiarize the animals with the treadmill. This acclimatization phase is carried out daily for two consecutive days, lasting around 10 minutes to ensure a smooth transition for the animals.

**Table 1. Parameters for the treadmill during the acclimatization period.**

|  | Speed (m/min) | Acceleration time (s) | Velocity duration (min) |
|---|---|---|---|
| Initial velocity | 5 | 5 | 3 |
| First-order velocity | 10 | 5 | 3 |
| Second-order velocity | 15 | 5 | 2 |

Table 2 outlines the parameter configurations for the animal experimental treadmill utilized during the testing phase. The stimulation current of the experiment is consistent with that of the acclimatization period. Mice are deemed to have reached exhaustion when they cease running for a continuous period of 10 seconds, at which point their endurance time is documented.

## Hematoxylin–Eosin (H & E) staining for GAS

After being immersed in 4% paraformaldehyde for more than 48 hours, the GAS tissue was meticulously trimmed, followed by a series of graded dehydration steps to produce paraffin-embedded blocks. The thickness was set to 5 μm and paraffin-embedded blocks was continuously sliced using a paraffin microtome (RM2255, Leica, Germany). The tissue sections were subsequently left to incubate in an oven set at a temperature of 60°C for the entire night. The paraffin-embedded sections were meticulously stained with the H & E staining kit (G1120, Solarbio, China) and subsequently examined under a Leica optical microscope (CTR6, Germany). The cross-sectional area of the muscle cells can be measured by analyzing the stained images of the tissue sections with Image J software (Version 1.8.0).

## RNA sequencing (RNA-Seq)

First, mouse skeletal muscle samples frozen with liquid nitrogen were transferred to a grinding tube, and Trizol (CW0580S, Cwbio, China) cracking buffer was added, ground by a grinder for 30 seconds, and then left for 5 minutes to achieve complete tissue lysis. Samples were then purified by centrifugation, chloroform/isoamyl alcohol (24:1) extraction, isopropyl alcohol precipitation, 75% ethanol washing, and short centrifugation. The precipitate was refrigerated at −20°C for 2 hours to facilitate the precipitation of RNA. Finally, the supernatant was removed, the precipitate was washed with 75% ethanol, followed by a brief centrifugation. Then, a pipette was used to remove the residual liquid. After drying for 3–5 minutes, the RNA precipitate was finally dissolved with 20–200 µl of DEPC-treated water or RNase-free water. The RNA was quantified and assessed using the Agilent 2100 bioanalyzer, followed by mRNA isolation with magnetic beads and subsequent high-temperature cleavage. Next, the mRNA molecules are fragmented to create shorter RNA pieces, which facilitates subsequent steps in the library preparation process. The fragmented mRNAs serve as templates for cDNA synthesis, which involves creating both the first-strand and second-strand cDNA through a series of enzymatic reactions. The double-stranded cDNA is then subjected to end repair to create blunt ends, followed by the addition of a single 'A' nucleotide to the 3' ends, which is necessary for adaptor ligation. After adaptor ligation, a PCR reaction is performed to amplify the cDNA fragments and prepare them for sequencing. After adaptor ligation, a PCR reaction is performed to amplify the cDNA fragments and prepare them for sequencing. Library QC is conducted again to ensure the quality of the amplified products before they are subjected to circularization. During circularization, single-stranded PCR products are cyclized to form circular DNA molecules, while any remaining linear DNA is digested. Finally, single-stranded circular DNA molecules undergo replication through a rolling circle amplification process, resulting in the formation of DNA nanoballs (DNBs). These DNBs contain multiple copies of the target DNA sequence. Following this, high-quality DNA was efficiently deposited into prefabricated nanoarrays and then sequenced using cPAS technology.

**Table 2. Parameters for the treadmill during the testing phase.**

|  | speed (m/min) | Acceleration time (s) | Velocity duration (min) |
| --- | --- | --- | --- |
| Initial velocity | 5 | 5 | 1 |
| First-order velocity | 10 | 5 | 2 |
| Second-order velocity | 20 | 5 | exhaustion |

Raw data were cleaned with SOA Pnuke v1.5.2, removing low-quality reads, adapter contamination, 'N' bases > 10%, and bases ≤ 15 quality > 50%. After filtering, clean reads were saved in FASTQ. The clean data were mapped to the reference genome by HISAT (v2.1.0) and the assembled unique gene by Bowtie2 (v2.2.5), and RSEM (v1.2.8) software was used to estimate the expression level of genes. Annotating assembled Unigene with seven major functional data-bases (KEGG, GO, NR, NT, SwissProt, Pfam, and KOG), and predicting the transcription factors. Time Series analy-sis was performed by Mfuzz (v2.34.0), and gene co-expression network analysis was performed by WGCNA (v1.48). Within-group differential gene analysis was performed using DESeq under the conditions of Fold Change ≥ 2 and Adjusted P value ≤ 0.001. PossionDis was performed between-group differential gene analysis under the conditions of Fold Change ≥ 2 and FDR ≤ 0.001.

According to the GO and KEGG annotation results and classifications, the differentially expressed genes were func-tionally classified, the phyper (https://en.wikipedia.org/wiki/Hypergeometric_distribution) in R software was used for KEGG enrichment analysis, and the TermFinder package was used for GO Enrichment analysis (https://metacpan.org/pod/GO::TermFinder). With a FDR of ≤ 0.05 as the threshold, candidate genes that met this condition were defined as signifi-cantly enriched. To identify DEGs, a series of analyses were performed, including clustering and functional enrichment assessments. The PoissonDis algorithm was utilized for DEG detection, and statistical significance was determined by the false discovery rate (FDR). For this study, DEGs were characterized as those with an FDR of ≤ 0.001 and a fold change of 2 or greater. Finally, graphical visualization of the data was performed using ggplot2 (v 3.4.3).

## Quantitative real-time PCR

Total mouse muscle RNA was extracted from frozen samples using Trizol reagent. Reverse transcription and fluorescence quantitative analysis were conducted with reagents (Cat No. 11141ES60 and Cat No. 11202ES08, Yeasen, Shanghai, China) and Real-time PCR instrument (ABI 7500). The primers used in the process were synthesized by Shenzhen BGI Co., LTD (Supplemental Appendix 1)

## Statistical analysis

Quantitative analysis of pathological staining was conducted with Image J software (version 1.8.0), and the stained area of each pathological section was pixelated. AOI tool was used to manually outline all the complete muscle fibers, remove the incomplete muscle fibers at the edge of the image, and measure the area of each muscle fiber within the AOI range, while data visualization and statistical analysis were handled by GraphPad Prism software (version 9.3.0). Data normality was evaluated using the Shapiro–Wilk test. For two-group comparison, if data follow a normal distribution, two-tailed Student's t-test was used. Nonparametric Wilcoxon rank sum test was used for non-normally distributed data. For comparisons in more than two groups, if data follow a normal distribution and meet the assumption of homogeneity of variances, analysis of one-way ANOVA followed by Tukey's post hoc test was used to compare group means. Nonparametric Kruskal-Wallis test with Dunn's multiple comparison test was used for non-normally distributed data. All statistical analyses were per-formed using R or GraphPad Prism 8, and differences of $P < 0.05$ were considered to be statistically significant. $*P < 0.05$, $**P < 0.01$, $***P < 0.001$.

## Result

### Lactoferrin and CoQ10 increased muscular mass

Male C57BL/6 mice were continuously injected with D-gal for 8 weeks to establish a murine sarcopenia mode. Mean-while, different groups were given the same amount of physiological saline, lactoferrin and CoQ10 solutions, is given daily through gavage feeding (Fig 1A). The weight change curve of mice indicated that the body weight of mice in each groups showed an upward tendency (Fig 1B). The results of mouse body composition analysis that the Model group's lean

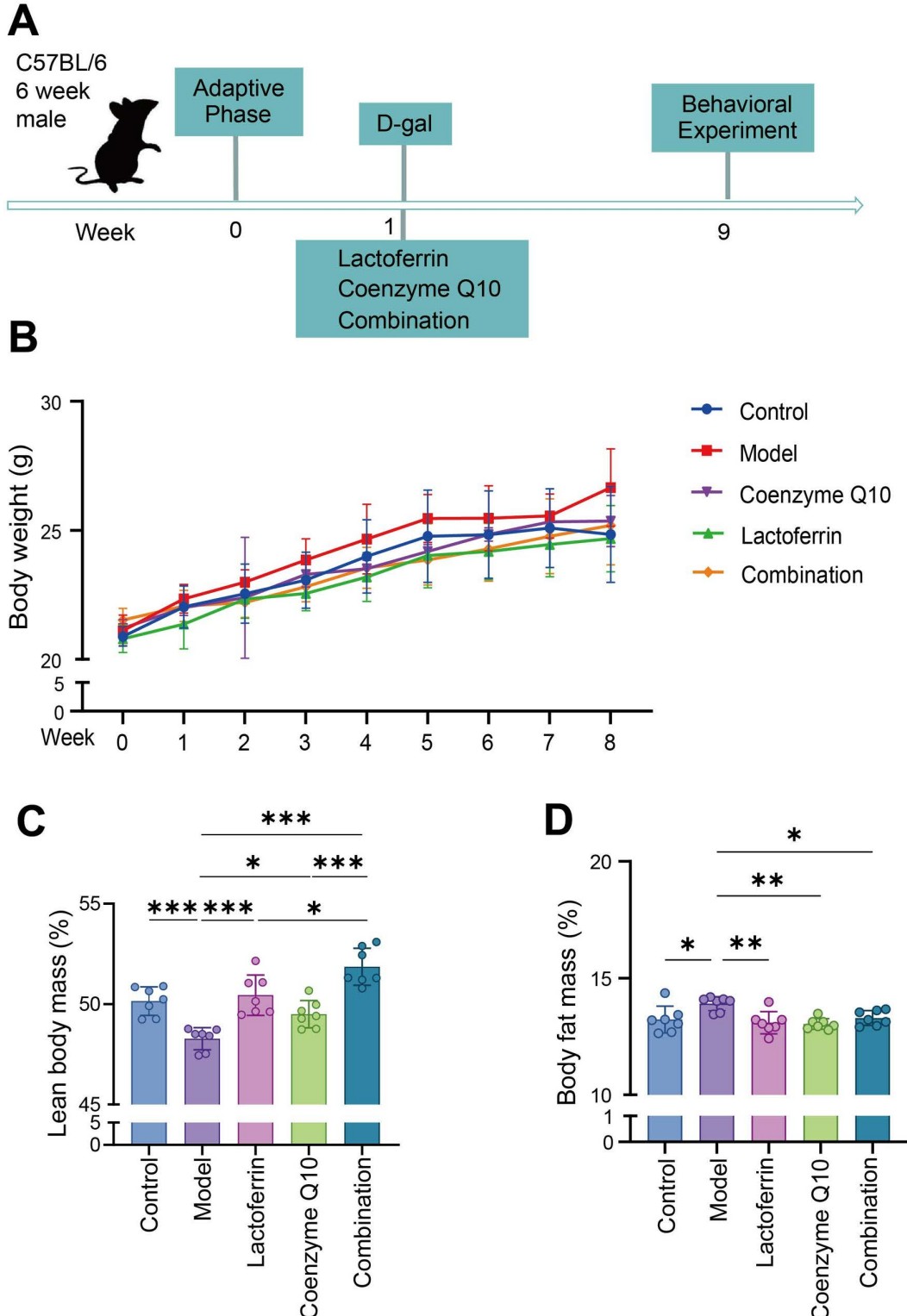

**Fig 1. Improvement effect of lactoferrin and Coenzyme Q10 on body composition of mice with sarcopenia.** (A) Experimental timeline. (B) Body weight. (C) Lean body content. Lean meat mass per mouse body weight * 100%. (D) Body fat content. Fat mass per mouse body weight. Results were presented as mean ± standard deviation. Each dot represents a mouse sample. Seven animals per group. Statistically significant differences were determined by one-way ANOVA and Tukey's post hoc test between groups (*$P < 0.05$, **$P < 0.01$, ***$P < 0.001$).

muscle mass was remarkably decreased, The lean meat content in the lactoferrin group, the coenzyme Q10 group, and the combined group all improved to different degrees (Fig 1C), among which the combined group was increased by the highest 2.273~5.365%. Moreover, the fat content of the Model group was higher than that of other groups (Fig 1D), with a decrease of 0.359~1.058% in the combined group, indicating that D-Gal intervention may lead to fat accumulation. In a word, this evinced that the percentage of muscular tissue in mice increased subsequent to the intervention. After the mice were euthanized, these skeletal muscle mass and the skeletal muscle index indicated that D-gal exacerbated the loss of skeletal muscle. The Model group had the lowest GAS quality, GAS muscle index and TA muscle index (Fig 2A, B, F). After the intervention, improvement was shown by all three groups and the most significant improvement was shown by the combination group (Fig 2A, B), where the GAS index increased by 0.259~0.335%. The quality changes of EDL, TA and SOL are negligible (Fig 2C, E, G) and the rest of skeletal muscle index is the same (Fig 2D, H). This may be due to the low proportion of these muscles in the total thigh muscles. These results indicate that lactoferrin and coenzyme Q10, either alone or in combination, can retard the decrease of skeletal muscle in mice. Moreover, the synergistic effect of the combination yields the most pronounced impact.

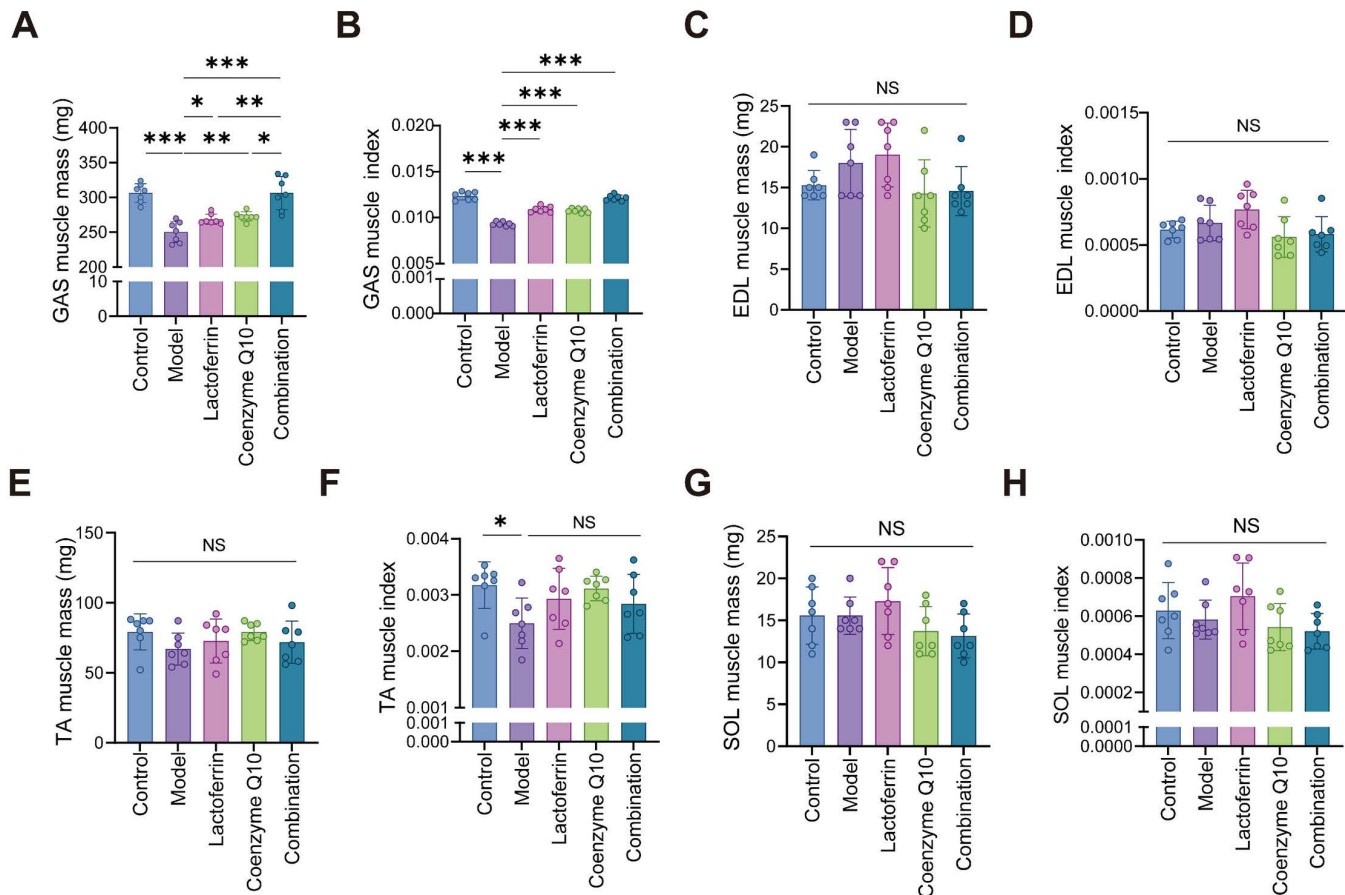

**Fig 2. Effects of lactoferrin and Coenzyme Q10 on skeletal muscle of mice.** (A) GAS muscular mass. (B) GAS muscle index. Ratio of mouse GAS muscular mass versus body weight. (C) EDL muscular mass. (D) EDL muscle index. (E) TA muscular mass. (F) TA muscle index. (E) SOL muscular mass. (F) SOL muscle index. The skeletal muscle index is the ratio of skeletal muscle to body weight. Seven animals per group. Statistically significant differences were determined by one-way ANOVA and Tukey's post hoc test between groups (NS $P > 0.05$, *$P < 0.05$, **$P < 0.01$, ***$P < 0.001$).

## Lactoferrin and CoQ10 ameliorated muscular strength and function

Sarcopenia results in reduced muscular strength and function. Therefore, the strength of the grip (Grip strength test), inverted suspension time (Inverted grid test) and exhaustion time (Endurance test) were measured in each group to assess muscular strength and function in mice. The, the maximum limb muscular strength value (grams) decreased significantly in the model group. However, treatment with Lactoferrin or CoQ10 prevented their reduction after the intervention. The Combination group indicated a more beneficial effect, with maximum limb muscle strength increasing by 6.83 to 53.498 g, dramatically outstripped the performance of the single-treatment arm (Fig 3A), denoting an amelioration in the muscular strength. Furthermore, the duration of inverted suspension and the time to exhaustion were both found to exhibit a positive correlation with muscular endurance. The findings revealed that both the inverted suspension duration and the exhaustion time shared an analogous trajectory in their respective changes (Fig 3B, C), Specifically, the inverted suspension time in the combined group was extended by 563–859 s, and the fatigue time was increased by 386–468 s. This denoted that muscle stamina can be remarkably boosted and thereby improve muscle function after single or combined intervention.

## Lactoferrin and CoQ10 improved the morphology of muscular tissue

We conducted a histological examination of the gastrocnemius muscle in mice by staining the tissue sections with hematoxylin-eosin staining (H & E). This staining technique enabled us to observe the pathological status of the muscle fibers, thereby facilitating an assessment of the alterations in the mouse muscle tissue. In normal mouse skeletal muscle sections, the cell membranes were distinctly defined, the transverse section of the muscular fibers exhibited a polygonal shape, and were neatly aligned. The fibers maintained a uniform diameter and size, with nuclei positioned at the periphery of the cells. The skeletal muscle fibers exhibited signs of disarray, with local loosening and a noticeable widening of the interstitial spaces between the muscle fibers in Model.

In contrast to the Model, the muscle cells in the Combination group were notably larger, displayed orderly cellular alignment, and had diminished interstitial elements, the muscle cells in the Combination group (Fig 4A–E). The analysis of the cross-sectional area (CSA) of muscle fibers revealed that the CSA of muscle cells in the model group was reduced. Upon supplementation with lactoferrin and coenzyme Q10, the CSA of muscle cells was restored, with the combination group showing a more significant improvement. The CSA of muscle cells exhibited no significant variations when lactoferrin and CoQ10 were administered, as compared to the Control (Fig 4F).

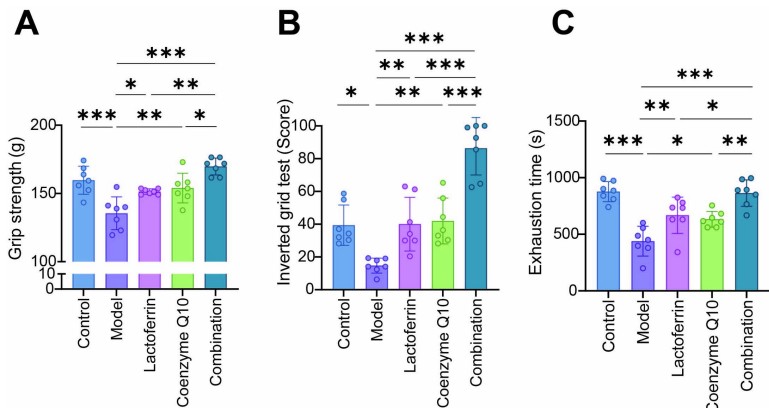

**Fig 3. Each group's performance on behavioral tests.** (A) The test of grip strength in mouse limbs. (B) The inverted hanging time of mice on a grid. (C) The exhaustion time of each group of mice in treadmill test. The results were presented as mean ± standard deviation. Seven animals per group. Statistically significant differences were determined by one-way ANOVA and Tukey's post hoc test between groups (*$P < 0.05$, **$P < 0.01$, ***$P < 0.001$).

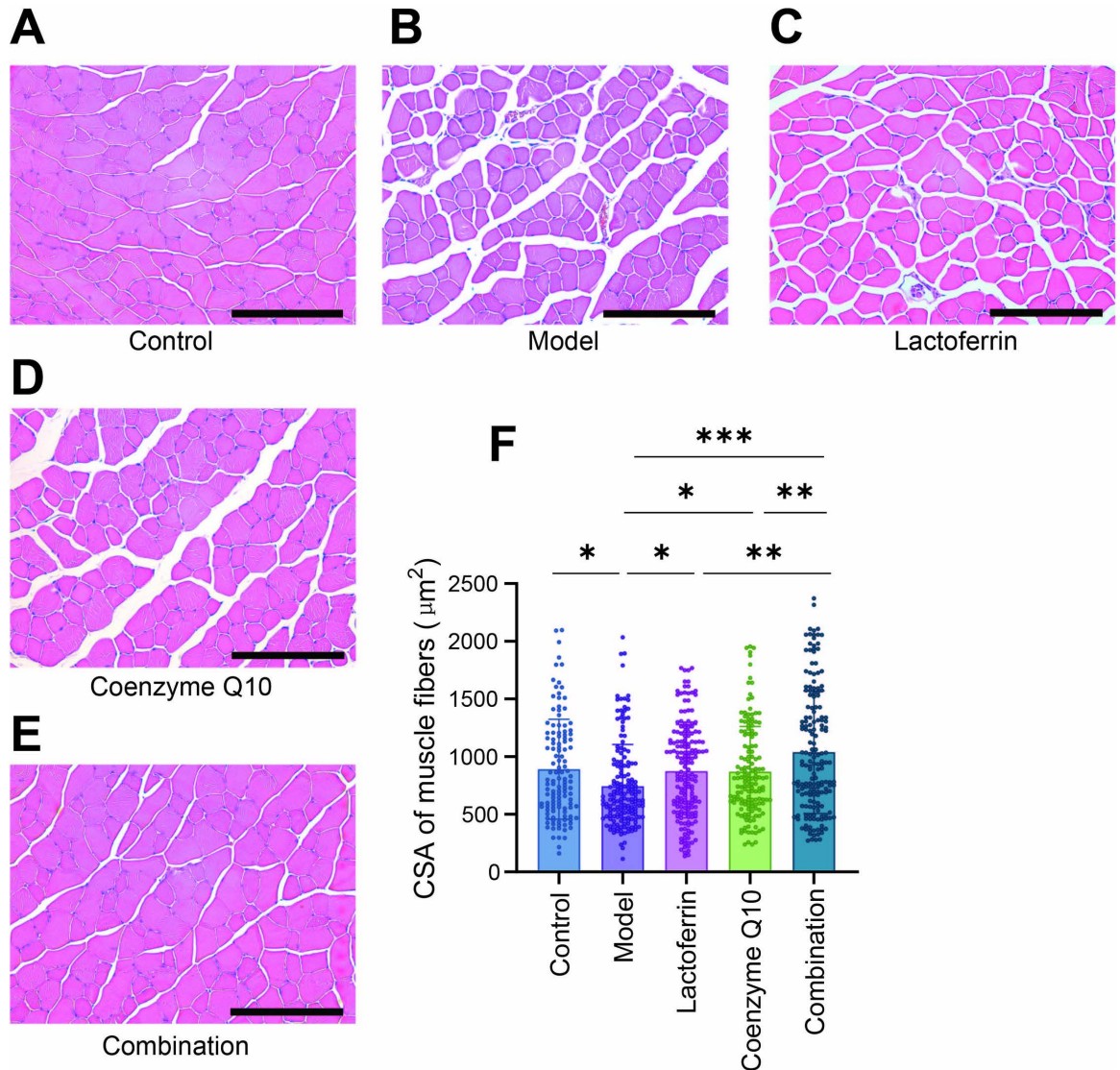

**Fig 4. HE staining of the gastrocnemius muscle.** (A) Control group; (B) Model group; (C) Lactoferrin group; (D) Coenzyme Q10 group; (E) Combination group. Each image is representative of a typical GAS muscle. Scale bar: 200 μm. (F) CSA of muscular fibers. Statistically significant differences were determined by one-way ANOVA and Tukey's post hoc test between groups (*$P < 0.05$, **$P < 0.01$, ***$P < 0.001$).

### RNA-seq in mice muscle tissue

In this research, we utilized the GEO database to identify DEGs. The discovery and subsequent analysis of these DEGs are instrumental in uncovering the underlying mechanisms associated with sarcopenia [34].

### Enrichment analysis of Gene Ontology (GO) annotations

The group that received CoQ10 supplementation exhibited a marked enrichment in biological processes when contrasted with the Model group. It encompasses the fundamental functions of cells, including cell activation and metabolism, cell development and differentiation, macromolecular biosynthesis, and nucleic acid metabolism and transcription. Collectively,

these processes support cell survival, growth, reproduction, and the execution of cellular functions (Fig 5A). GO analysis revealed that the majority of DEGs were predominantly linked to molecular functions, especially those related to cation, protein, and catalytic activities, as well as bindings to heterocyclic compounds, carbohydrates, purine nucleosides, ribonucleosides, and kinase activities involving small molecules (Fig 5D). The findings suggest that CoQ10 primarily exerts its effects on cellular biosynthesis and metabolic pathways. As a key electron carrier in the mitochondrial respiratory chain, CoQ10 is involved in oxidative phosphorylation, which helps convert chemical energy from nutrients into ATP, a form of energy available to cells. This process is at the heart of cellular metabolism, ensuring that cells are able to use energy efficiently and maintain normal life activities. Secondly, the synthesis process of CoQ10 is itself an important part of cell biosynthesis, and its synthesis involves a variety of enzymes and metabolic intermediates, such as p-hydroxybenzoic acid (4-HB) [35] as the precursor of the CoQ10 head group, which are involved in complex biosynthetic pathways. This direct involvement in biosynthetic pathways and indirect regulation of metabolic processes reinforces the central role of CoQ10 in cell biosynthesis and metabolism. Compared to the Model group, the GO results showed that the Lactoferrin group was significantly enriched for various biological processes, including cell activation, metabolism, macromolecule metabolism, development, cell differentiation, movement, biosynthesis, cytoskeleton organization, and organ development, particularly in the regulation of protein metabolism (Fig 5B). It is mainly linked to molecular functions such as catalytic activity, protein and ion binding, carbohydrate derivative and metal ion interactions, and bindings with purine nucleosides, ribonucleosides, and small molecules (Fig 5E). These findings indicated that lactoferrin may mainly have an impact on cellular differentiation, metabolism and biosynthesis.

After conducting a thorough comparison and analysis between the Model group and the Combination group, it was ascertained that five out of the top fifteen most enriched biological processes were also significantly enriched in the Lactoferrin group (locomotion, macromolecule biosynthetic process, cytoskeleton organization, cellular macromolecule biosynthetic process, animal organ development), and two processes were also significantly enriched in the CoQ10 group: cellular metabolic and developmental process. The Combination group specifically enriched regulation of locomotion, actin filament-based process, actin cytoskeleton organization and cell proliferation (Fig 5C). These results further reveal that lactoferrin and CoQ10 exhibit certain synergies and complementarity in regulating cellular biological processes. The biological processes that were significantly enriched in the lactoferrin group, such as motility and cytoskeletal organization, were highly consistent with the processes that were specifically enhanced in the combination group, suggesting that lactoferrin may play a dominant role in these processes.

Among the top ten enriched molecular functions, seven were found to be significantly enriched across all intervention groups (protein binding, catalytic activity, carbohydrate derivative binding, purine nucleoside binding, ribonucleoside binding, purine ribonucleoside binding, small molecule binding) and two showed significant enrichment in the Combination group (including organic cyclic compound binding and protein kinase activity) and CoQ10 group (including ion binding and metal ion binding), respectively (Fig 5F). The findings reveal that the Combination group exerts a comprehensive influence on cellular biosynthesis, development, differentiation, and metabolic processes. Furthermore, it exerts a substantial influence on both catalytic activity and kinase function. In summary, lactoferrin and CoQ10 demonstrate a significant synergistic effect in the regulation of cellular molecular functions. The significant enrichment of molecular functions in the combination group, such as organic cyclic compound binding and protein kinase activity, indicates that it exerts a comprehensive and profound influence on cellular biosynthesis, development, differentiation, and metabolism processes. This synergistic effect is not only evident in the fundamental metabolic functions of cells but also further promotes the optimization of physiological functions through enhanced catalytic activity and improved kinase function. For example, lactoferrin, with its potent iron-binding capacity, regulates iron metabolism within cells, thereby influencing cell proliferation and differentiation [36]. The significant enhancement of the binding function of organic cyclic compounds in the combination group may be attributed to the synergistic effects of lactoferrin and CoQ10 on cellular metabolism. Lactoferrin impacts cellular energy metabolism by modulating iron ion levels, while CoQ10, a crucial component of the cellular respiratory

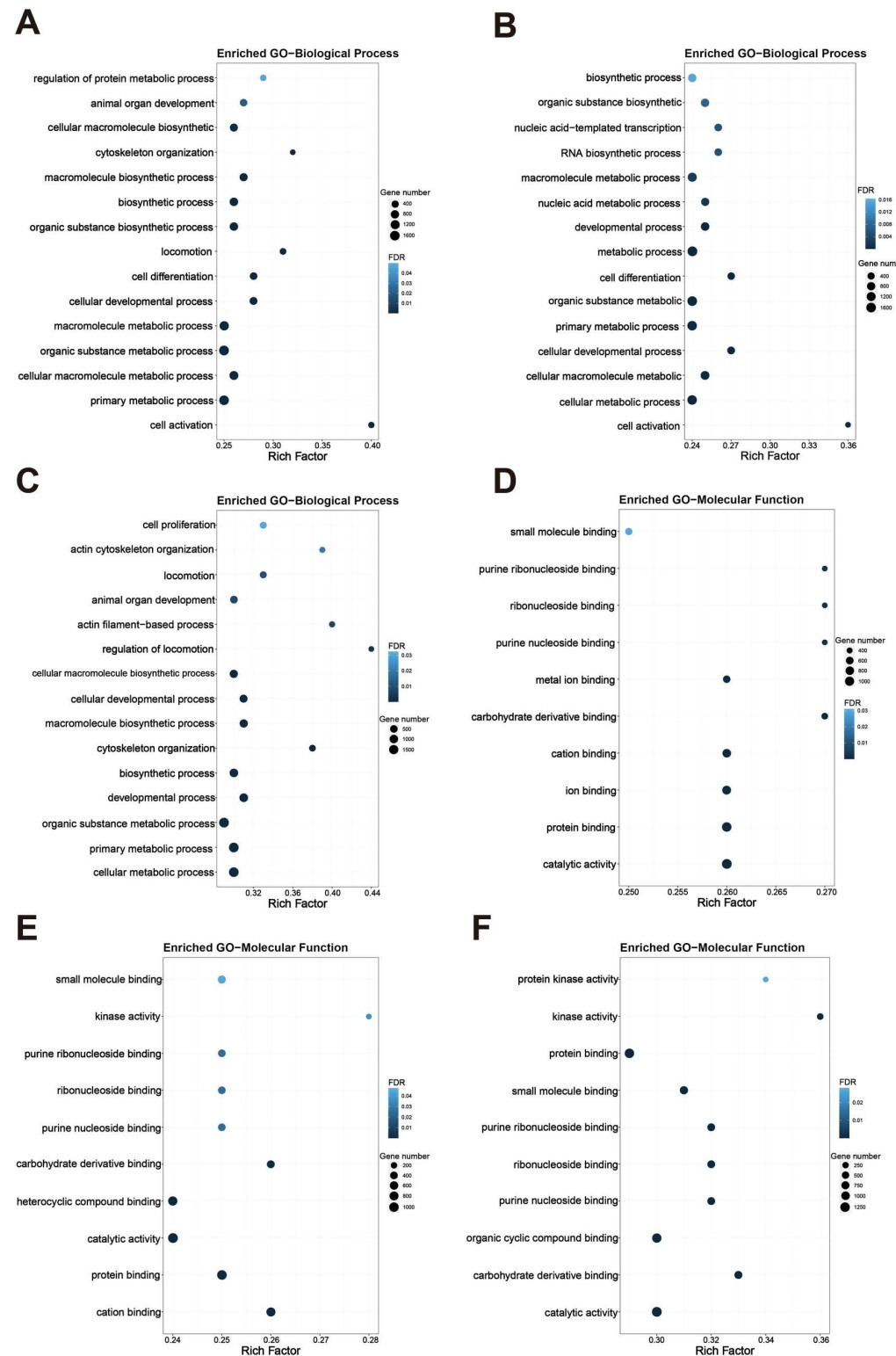

**Fig 5. Transcriptome sequencing enriched sarcopenia related GO Biological process (BP) and Molecular function (MF). Enriched GO BP of DEGs.** Model group was compared with Lactoferrin group (A), Coenzyme Q10 group (B), Combination group (C). Enriched GO MF of DEGs. Model was compared with Lactoferrin group (D), Coenzyme Q10 group (E), Combination group (F). The X-axis represents Rich Factor, and the Y-axis represents BP and MF names. The size of the point represents the number of DEGs. (A-F) FDR ≤ 0.05 is considered significant.

chain, facilitates energy production in cells. This synergistic interaction not only improves cellular metabolic efficiency but may also influence cell development and differentiation by regulating intracellular signaling pathways.

## Analysis of enriched KEGG signaling pathways

The KEGG pathway analysis revealed that the CoQ10 group was notably enriched in key biological processes, including DNA replication, cell cycle regulation, ECM-receptor interactions, sphingolipid signaling, steroid biosynthesis, pantothenate and CoA biosynthesis, CAMs, nucleotide excision repair, phosphonate metabolism, and the Rap1 signaling pathway (Fig 6A), while the Lactoferrin group was significantly enriched for the Protein digestion and absorption, Steroid biosynthesis, Cell adhesion molecules (CAMs), Hippo signaling pathway, ECM-receptor interaction, MAPK signaling pathway, Phosphatidylinositol signaling system, Focal adhesion, Cell cycle, Regulation of actin cytoskeleton (Fig 6B), compared to the Model. The Combination group was also significantly enriched for the ErbB signaling pathway, p53 signaling pathway, Hippo signaling pathway, Phosphatidylinositol signaling system, Cell cycle, AMPK signaling pathway, MAPK signaling pathway, DNA replication, Regulation of actin cytoskeleton, Wnt signaling pathway (Fig 6C). According to the KEGG pathway analysis results, the CoQ10 group and the lactoferrin group exhibited significant enrichment in several key biological processes, while the combined group further integrated these functions, demonstrating a broader impact. Both the CoQ10 group and the lactoferrin group exhibited significant enrichment in pathways associated with cell cycle regulation, DNA replication, and MAPK signaling. These pathways are fundamental mechanisms governing cell proliferation and differentiation. For instance, the MAPK signaling pathway facilitates the progression of the cell cycle by modulating the activity of various enzymes within the cell [36]. Furthermore, the combination group amplified the activity of these pathways, indicating a synergistic effect in promoting cell proliferation and differentiation. Both the lactoferrin group and the combination group were significantly enriched in pathways related to actin cytoskeleton regulation, which may be associated with enhanced muscle function. The dynamic changes in the actin cytoskeleton are essential for muscle contraction and motor function [36]. Additionally, the combination group showed significant enrichment in the Wnt signaling pathway, which plays a crucial role in muscle regeneration and repair. The outcomes demonstrated that the application of lactoferrin and CoQ10, whether individually or in tandem, correlates with enhanced cell proliferation, differentiation, and senescence, as well as improved muscle functionality.

## Validation of the mRNA expression levels of candidate genes by qPCR

Muscular fibers are the fundamental units of skeletal muscle and play a crucial role in muscle development, growth, repair, and regeneration. During aging, the number of muscular fibers may decrease, while the diameter of muscular fibers may also decrease, which affects the overall function and strength of the muscles [37]. *MyoD1, Myf5, Mef2c* and *Myog* are transcription factors closely associated with muscle fiber development. They play a crucial role in the regulation of muscle fiber formation and function [38–41]. The proteins encoded by *Myoz2* and *Myh2* are involved in muscle contraction and intracellular signaling, thus promoting the muscle fiber thickening [42,43], and the findings revealed that the mRNA expression levels of *MyoD1, Myf5, Mef2c, Myog, Myoz2* and *Myh2* were up-regulated in single or combined groups (Fig 7A, B). *Fgf9* has been demonstrated to plays a role in stimulating the activation and proliferation of muscle stem cells, which are vital for the healing and regeneration processes of muscles that have been damaged. In one study, muscles treated with *Fgf9* showed near-normal size muscular fibers and reduced interstitial fibrosis [37]. *Fgf9* (10 ng/mL) can increase the expression of *Icam-1*, and the upregulation of *Icam-1* enhances muscle adhesion and fusion, thereby facilitating the repair and regeneration of the injured muscle structure [44]. Furthermore, by improving the metabolic function of the mitochondria, *Sirt3* is able to maintain the energy metabolism of muscle cells and mitigate oxidative stress injury [45], and the results showed that the mRNA expression levels of *Fgf9, Icam-1* and *Sirt3* were up-regulated in single or combined groups (Fig 7C).

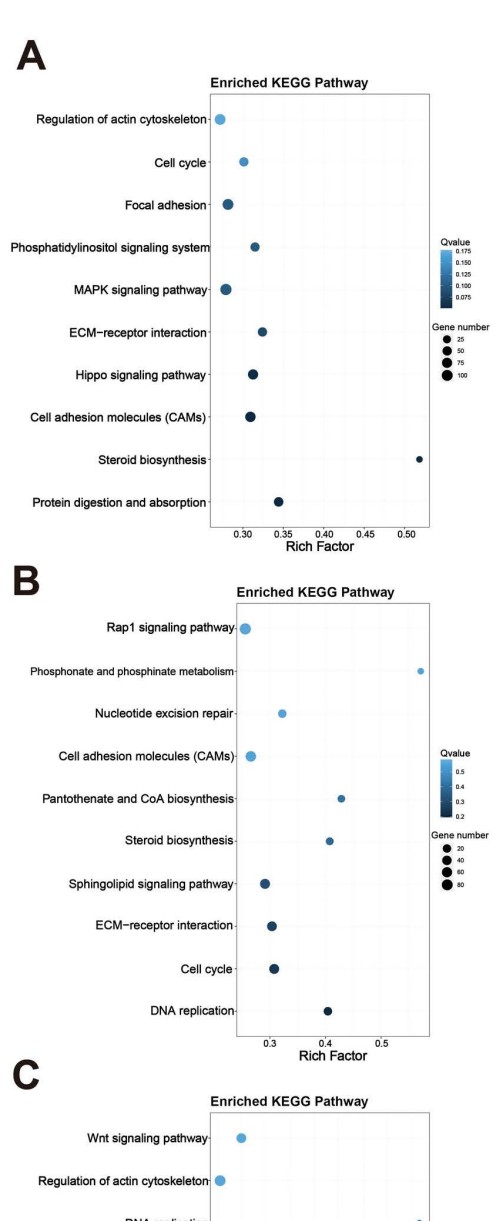

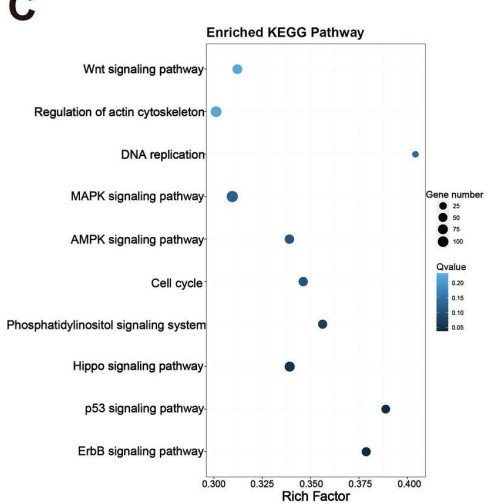

**Fig 6. KEGG pathway enrichment analysis of lactoferrin and Coenzyme Q10 in sarcopenia.** Model was compared with Lactoferrin group (A), Coenzyme Q10 group (B), Combination group (C). The X-axis represents Rich Factor, and the Y-axis represents pathway names. The colors represent Q values. The size of the point represents the number of DEGs.

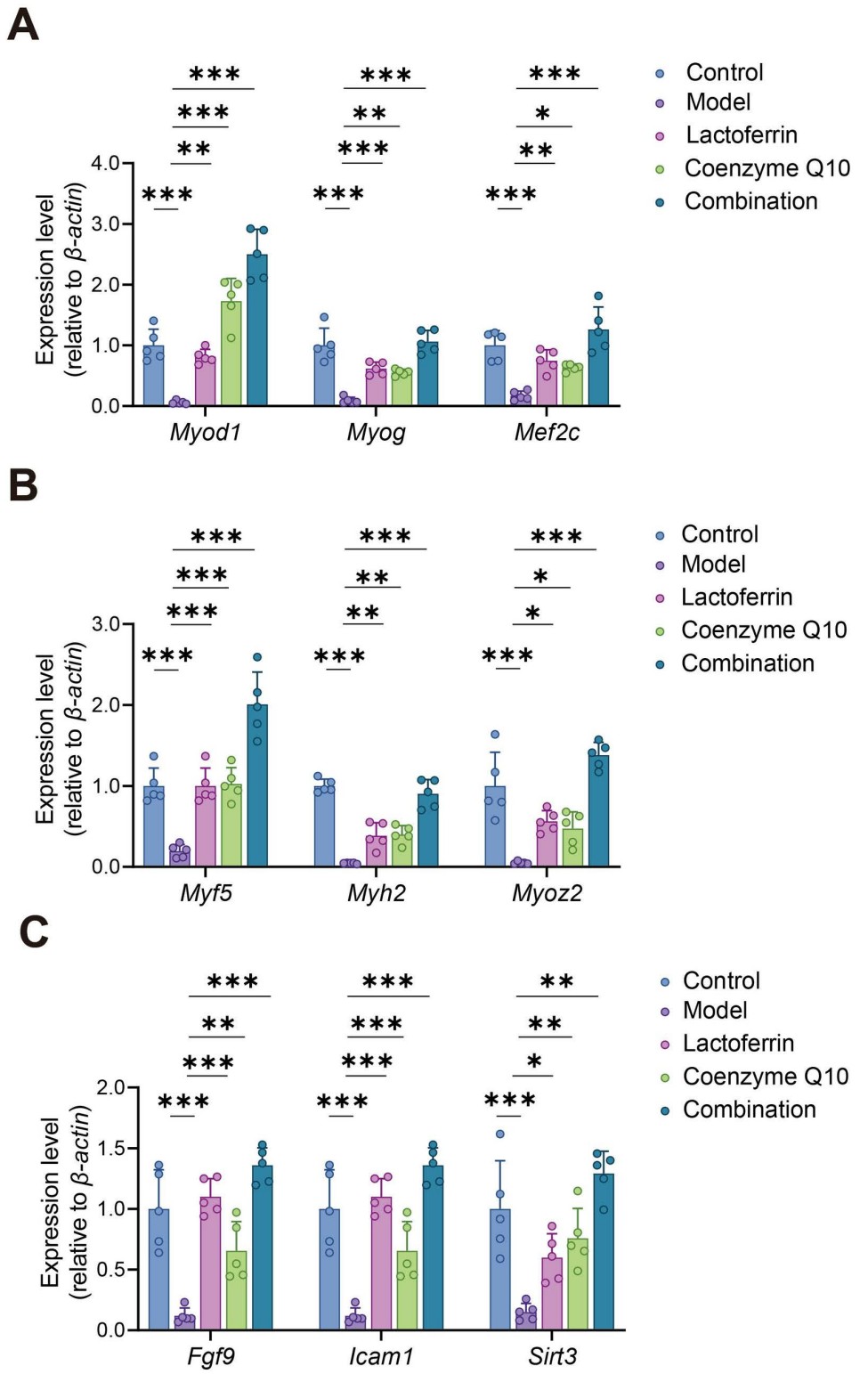

**Fig 7. qPCR verified the expression changes of GO and KEGG pathway enriched sarcopenia related DEGs.** The mRNA expression of (A) *Myod1*, *Myog,* and *Mef2c*; (B) *Myf5, Myh2,* and *Myoz2*; (C) *Fgf9, Icam1,* and *Sirt3*. Five animals per group. Statistically significant differences were determined by one-way ANOVA (*$P<0.05$, **$P<0.01$, ***$P<0.001$).

## Discussion

Sarcopenia is an age-related syndrome characterized by a progressive loss of muscular mass and strength, as well as a decline in muscle function. With aging, the human body is more susceptible to a variety of musculoskeletal diseases [46], and sarcopenia is one of them. In addition to age, allergic reactions (atopic dermatitis, asthma, etc.) can also cause sarcopenia [47]. According to the metabolic theory of aging, aging is the result of metabolic disorders in the body [48]. The disorder of glucose metabolism leads to the increase of free radical production, attacks the lipids and proteins located in the inner membrane of mitochondria, affects the function of mitochondria, and leads to cell damage, aging and disease. The D-gal aging model is similar to natural aging on the whole, and can be used to study the effects of anti-aging drugs in behavioral, antioxidant and other aspects on the whole level [49]. Therefore, in this study, D-gal induced mouse sarcopenia model was established. At present, the diagnosis and evaluation of sarcopenia primarily rely on the assessment of muscle mass, strength, and functionality. For a more comprehensive understanding, additional parameters such as muscle fiber composition and physical performance indicators are also considered.

Following eight weeks of D-Gal administration via intraperitoneal injection, there was a significant reduction in lean mass, gastrocnemius muscle mass (GAS mass), and the GAS index in the Model group as compared to the Control. Furthermore, measurements of grip strength, endurance, and suspension time all pointed towards a decline. Pathological staining revealed a decrease in muscle fiber cross-sectional area and an increase in intercellular spacing. These findings highlight a substantial decrease in muscular mass and function within the sarcopenia mice, confirming the efficacy of the D-galactose intervention model.

The supplementation with Lactoferrin or CoQ10 not only halted the progression of sarcopenia but also promoted muscle growth and recovery in the treated mice. The significant increase in lean mass, GAS mass, and GAS index underscores the potential of these interventions to enhance muscle health and function.

The improvement effect of the Lactoferrin group on lean meat content was better than that of the CoQ10 group. Muscular mass was significantly higher in the Combination group. According to the EWGSOP and related studies, muscular strength is considered more critical than muscle mass for diagnosing and assessing the risk of sarcopenia [3,50]. This shift in focus emphasizes the importance of muscle function over mere quantity, highlighting that low muscle strength is a key characteristic of sarcopenia and a significant indicator of its severity. The behavioral test outcomes from this study demonstrated notable enhancements in muscular strength and functionality within the intervention group, with particularly striking improvements observed in the Combination group. This suggests that the synergistic effects of the combined treatment may offer superior benefits in bolstering muscle performance compared to individual interventions. Pathological staining results agreed to confirm these results, and together these findings suggest that lactoferrin and CoQ10 intervention can effectively alleviate sarcopenia.

The mechanism by which lactoferrin and CoQ10 alleviate sarcopenia was investigated through RNA sequencing (RNA-seq). The RNA sequencing data revealed that all groups receiving the treatment were significantly enriched about the Cell cycle. Both lactoferrin and CoQ10 intervention groups enriched ECM-receptor interactions and cell adhesion molecules, which provided the basis for maintaining high cytoskeleton stability [51]. The RNA-seq analysis highlighted that the lactoferrin-supplemented group showed significant enrichment in the Focal Adhesion pathway, which plays a crucial role in cell migration and adhesion—both crucial processes for the repair of wounds and the restoration of injured muscle tissue [52]. This enrichment suggests that lactoferrin may facilitate muscle repair and recovery by enhancing these cellular activities. In addition, the CoQ10 group enriched the Rap1 signaling pathway, which is crucial for the proliferation and migration of vascular smooth muscle cells via the ERK signaling pathway. The findings imply that the intervention might influence muscle function by affecting the proliferation of muscle cells, protein metabolism, and cytoskeleton.

As mentioned earlier, cytoskeletal stability is also extremely important, and the significant enrichment of the regulation of actin cytoskeleton in the Lactoferrin and Combination group further confirms this conclusion. As previously mentioned, the activation and maintenance of skeletal muscle cell is crucial for muscle regeneration [53]. The maintenance of

cytoskeletal stability relies heavily on Cell adhesion molecules, ECM-receptor interaction, Regulation of actin cytoskeleton, and these KEGG pathways are significantly enriched in the Lactoferrin and Combination groups. Additionally, the KEGG enrichment results showed significant enrichment of the ErbB signaling pathway, p53 signaling pathway, AMPK signaling pathway, Wnt signaling pathway found only in the Combination group. N.k. Brasseur et al. confirmed the expression and localization of ErbB2, ErbB3 and ErbB4 proteins in skeletal muscle, and confirmed that the ErbB signaling pathway induced by contractile activity in vivo is related to the metabolic and proliferative response of skeletal muscle to exercise [54]. Current evidence has confirmed that p53 acts as a threshold regulator of cell homeostasis. By analyzing and determining the parameters (intensity and duration) of oxidative stress factors, the role of p53 signaling pathways to maintain cellular health and function of skeletal muscle is determined [55]. P. Scheale et al. believe that targeting the Wnt pathway is a promising approach for treating neuromuscular degenerative diseases [56]. In skeletal muscle, a large number of studies have brought to light the significance of AMPK as a mediator of cell signaling pathways that are closely associated with muscle function and metabolism, including glucose absorption, fatty acid oxidation, mitochondrial formation, protein metabolism, and muscle plasticity [57–59]. The results showed that the Combination group can improve mitochondrial function and protein metabolism in the muscles of sarcopenic mice, promote skeletal muscle cell proliferation and muscle regeneration, and was superior to the single intervention group.

In terms of combined use, previous studies have demonstrated that the combination of Coenzyme Q10 (CoQ10) and lactoferrin can mitigate hepatic and renal injuries induced by thioacetamide [30], and there are no obvious side effects at present. In the present study, we further found that the combination of CoQ10 and lactoferrin exhibited significant efficacy in a D-gal-induced mouse model. The inhibitory effect of lactoferrin on sarcopenia may impact skeletal muscle regeneration by enhancing protein digestion and absorption. CoQ10 may affect energy metabolism in skeletal muscle by improving mitochondrial function. The analysis showed that the combined treatment group was more effective in promoting protein metabolism and mitochondrial energy metabolism in the mice, which positively influenced the regeneration of muscular tissue. Although there are notable differences in physiological and metabolic mechanisms between mice and humans, the aging characteristics of the D-gal mouse model still share similarities with human aging. When gavaged to mice, lactoferrin and coenzyme Q10 did not specifically target muscle cells and might affect other tissues. potentially causing off-target effects. However, the current study suggests that there are no significant adverse effects associated with either compound. Therefore, the promising results observed in the D-gal mouse model suggest that this combined treatment holds potential for human applications, especially for vulnerable populations such as the elderly and patients with chronic diseases, and lays the foundation for further clinical research and application.

## Supporting information

**S1 Table.  Contents of Supplemental Materials.**
(DOCX)

## Author contributions

**Conceptualization:** Yongting Luo.

**Funding acquisition:** Yongting Luo.

**Investigation:** Wenbin Wu, Yinhua Zhu, Yanan Fu, Hongfei Xing, Xinlu Guo, Jichao Xu, Wenhui Hu, Mingyang Cui, Jiaxin Shi, Ling Li, Weiwei Wang.

**Project administration:** Yongting Luo, Junjie Luo, Qingchang Xing.

**Supervision:** Yongting Luo, Junjie Luo, Qingchang Xing.

**Visualization:** Wenbin Wu, Yinhua Zhu, Yanan Fu, Hongfei Xing, Xinlu Guo, Ling Li, Weiwei Wang.

**Writing – original draft:** Wenbin Wu, Yinhua Zhu, Yanan Fu.

**Writing – review & editing:** Peng An, Yongting Luo, Junjie Luo, Qingchang Xing.

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
