## [Decision Letter · Decision Letter 0]

5 Feb 2025

Dear Dr. Luo,

Thank you for submitting your manuscript to PLOS ONE. After careful consideration, we feel that it has merit but does not fully meet PLOS ONE’s publication criteria as it currently stands. Therefore, we invite you to submit a revised version of the manuscript that addresses the points raised during the review process.

https://journals.plos.org/plosone/s/submission-guidelines#loc-laboratory-protocols . Additionally, PLOS ONE offers an option for publishing peer-reviewed Lab Protocol articles, which describe protocols hosted on protocols.io. Read more information on sharing protocols at https://plos.org/protocols?utm_medium=editorial-email&utm_source=authorletters&utm_campaign=protocols .

We look forward to receiving your revised manuscript.

Kind regards,

Zahra Khodabandeh

Academic Editor

PLOS ONE

Journal Requirements:

2. To comply with PLOS ONE submissions requirements, in your Methods section, please provide additional information regarding the experiments involving animals and ensure you have included details on (1) methods of sacrifice, and (2) efforts to alleviate suffering.

“This work was supported by the National Key Research and Development Program of China (2023YFF1103501).”

Reviewers' comments:

Reviewer's Responses to Questions

**Comments to the Author**

1. Is the manuscript technically sound, and do the data support the conclusions?

Reviewer #1: Yes

Reviewer #2: Partly

2. Has the statistical analysis been performed appropriately and rigorously?

Reviewer #1: Yes

Reviewer #2: I Don't Know

3. Have the authors made all data underlying the findings in their manuscript fully available?

Reviewer #1: Yes

Reviewer #2: Yes

4. Is the manuscript presented in an intelligible fashion and written in standard English?

Reviewer #1: Yes

Reviewer #2: Yes

Reviewer #1: The introduction is comprehensive and well-written, however, it could be more concise in some areas, particularly in the discussion of CoQ10's antioxidant properties, which is somewhat repetitive. It should be completed by adding some more recent studies about its ebneficial effects on other aspects of human health. Also write same for lactoferin and also their combination. Use the below studies to complete it:

Role of Coenzyme Q10 in Health and Disease: An Update on the Last 10 Years (2010–2020)

Effect of Sunset Yellow on Testis: Molecular Evaluation, and Protective Role of Coenzyme Q10 in Male Sprague-Dawley Rats

Lactoferrin: A glycoprotein that plays an active role in human health

Chapter Six - Beneficial antioxidant effects of Coenzyme Q10 on reproduction

Method:

Add the weight of animals.

Add a reference for the dosage of each substance used.

Why 8 weeks selected for intervention?

The methods section is detailed and well-organized, providing sufficient information for replication.

The use of a D-galactose-induced aging model is appropriate, and the experimental design is sound.

The description of the RNA-seq analysis could be more detailed, particularly in terms of the bioinformatics pipeline and the criteria used for identifying differentially expressed genes (DEGs).

The statistical analysis section could be expanded to include more details on the specific tests used and the rationale for choosing them.

Results:

The results are presented clearly and logically, with appropriate use of figures and tables to support the findings. The data on muscle mass, strength, and function are compelling, and the RNA-seq analysis provides valuable insights into the underlying mechanisms.

The discussion is thorough and well-reasoned, effectively interpreting the results in the context of existing literature.

The discussion could also benefit from a more forward-looking perspective, suggesting future research directions or potential clinical applications.

Reviewer #2: The abstract could provide more specific details about the magnitude of improvements observed, particularly in the combined treatment group, to give readers a clearer sense of the study's impact.

The keywords are relevant.

Introduction:

The section could also benefit from a clearer statement of the study's novelty or how it advances existing knowledge in the field.

The methods section is detailed but could be improved in several areas:

The ethical statement is clear but could be more specific about the measures taken to minimize animal suffering, such as details on anesthesia or euthanasia protocols.

The RNA-seq analysis lacks sufficient detail, particularly regarding the bioinformatics pipeline and the criteria used for identifying differentially expressed genes (DEGs).

The results are presented clearly but could benefit from a more detailed discussion of the statistical significance, particularly in the context of the RNA-seq data. Some figures, while well-designed, have legends that could be more informative, particularly in explaining the significance of the findings.

The authors could provide a more critical evaluation of the study's limitations, such as the generalizability of the findings to humans or the potential for off-target effects of the interventions in the discussion.

The references are generally appropriate but could be more up-to-date, particularly in the discussion of the mechanisms of sarcopenia and the role of nutritional interventions. There are also inconsistencies in formatting, such as the use of italics and capitalization, which should be standardized.

**Do you want your identity to be public for this peer review?** For information about this choice, including consent withdrawal, please see our Privacy Policy

Reviewer #1: No

Reviewer #2: No

---

## [Author Response · Author response to Decision Letter 1]

20 Feb 2025

Letter to the Editor

Dear Dr. Zahra Khodabandeh,

We would like to sincerely thank you for sending out our manuscript for peer review. We also fully appreciate the constructive and insightful comments from the two outside experts; they helped to greatly improve our manuscript. To address all points raised by the Reviewers, we reanalyzed our data and inserted additional text. We also revised the manuscript according to Journal Requirements. Revisions and changes to the manuscript are clearly highlighted. Enclosed, please find our revised manuscript and a point-by-point response to address all the concerns raised by the Reviewers.

We hope the revised manuscript is of sufficient merit to be reconsidered for publication in the PLOS One. We look forward to your response in due course.

Best regards,

Sincerely,

Yongting Luo, PhD.

China Agricultural University

No. 17 Qinghua East Road, Haidian District,

Beijing 100193, China

E-mail: luo.yongting@cau.edu.cn

Point-by-point response

We would like to sincerely thank all the reviewers for their careful and critical review of our manuscript. We are also grateful for the encouraging comments and insightful suggestions that allowed us to substantially improve our manuscript. The following is our point-by-point response.

Reviewer #1:

Comment 1�The introduction is comprehensive and well-written, however, it could be more concise in some areas, particularly in the discussion of CoQ10's antioxidant properties, which is somewhat repetitive. It should be completed by adding some more recent studies about its beneficial effects on other aspects of human health. Also write same for lactoferrin and also their combination. Use the below studies to complete it:

1.Role of Coenzyme Q10 in Health and Disease: An Update on the Last 10 Years (2010–2020).

2.Effect of Sunset Yellow on Testis: Molecular Evaluation, and Protective Role of Coenzyme Q10 in Male Sprague-Dawley Rats.

3.Lactoferrin: A glycoprotein that plays an active role in human health.

4.Chapter Six - Beneficial antioxidant effects of Coenzyme Q10 on reproduction.

Response 1�Thank you for your constructive comment. As suggested, the discussion of CoQ10's antioxidant properties was shortened (page 4 lines 65-71), and its other beneficial effects was added (page 3 line 61-65). In addition, we also added additional text for lactoferrin and their combination by citing the above references (page 4 lines 78-92).

Comment 2�Method: Add the weight of animals.

Response 2�As suggested, we added the weight of animals in Material and Methods (page 6 line 14).

Comment 3: Add a reference for the dosage of each substance used.

Response 3: The CoQ10 and lactoferrin doses are 10 mg/kg/day and 500 mg/kg/day, respectively. According to previous studies [1-2] and the Meeh-Rubner formula, the dosages of CoQ10 and lactoferrin selected in this paper are within the range of effective dosages that the human body can ingest. We have added the two references into the revised manuscript (page 6 lines 22-23) to make it clear.

1. Andalib S, Shayanfar A, Khorrami A, Maleki-Dijazi N, Garjani A. Atorvastatin reduces the myocardial content of coenzyme Q10 in isoproterenol-induced heart failure in rats. Drug Res (Stuttg). 2014;64(5):246-250. doi:10.1055/s-0033-1357178

2. Wu W, Guo X, Qu T, et al. The Combination of Lactoferrin and Creatine Ameliorates Muscle Decay in a Sarcopenia Murine Model. Nutrients. 2024;16(12):1958. Published 2024 Jun 19. doi:10.3390/nu16121958

Comment 4�Why 8 weeks selected for intervention?

Response 4�It typically takes 8 weeks for D-gal to induce muscle aging in C57BL6 mice [1-2]�including loss of the muscle mass, reduced muscle strength and muscle cell atrophy. Therefore, 8 weeks was selected for interventions. We added these references into the revised manuscript (page 6 line 22) to make it clear.

1. Wang HH, Zhang Y, Qu TQ, et al. Nobiletin Improves D-Galactose-Induced Aging Mice Skeletal Muscle Atrophy by Regulating Protein Homeostasis. Nutrients. 2023;15(8):1801. Published 2023 Apr 7. doi:10.3390/nu15081801

2. Wu W, Guo X, Qu T, et al. The Combination of Lactoferrin and Creatine Ameliorates Muscle Decay in a Sarcopenia Murine Model. Nutrients. 2024;16(12):1958. Published 2024 Jun 19. doi:10.3390/nu16121958

Comment 5�The methods section is detailed and well-organized, providing sufficient information for replication. The use of a D-galactose-induced aging model is appropriate, and the experimental design is sound. The description of the RNA-seq analysis could be more detailed, particularly in terms of the bioinformatics pipeline and the criteria used for identifying differentially expressed genes (DEGs).

Response 5: Thank you for your positive and constructive comment. As suggested, the description of the bioinformatics pipeline and the criteria for RNA-seq analysis has been refined and updated in detail (page 10 lines 05-17).

Comment 6�The statistical analysis section could be expanded to include more details on the specific tests used and the rationale for choosing them.

Response 6: Thank you for your constructive comment. The Statistical Analysis section has been expanded to include more details about the specific tests used and the reasons for choosing them (page 11 lines 35-46).

Comment 7�Results: The results are presented clearly and logically, with appropriate use of figures and tables to support the findings.

The data on muscle mass, strength, and function are compelling, and the RNA-seq analysis provides valuable insights into the underlying mechanisms.

The discussion is thorough and well-reasoned, effectively interpreting the results in the context of existing literature.

The discussion could also benefit from a more forward-looking perspective, suggesting future research directions or potential clinical applications.

Response 7: Thank you very much for your positive and constructive comments. As suggested, we have added relevant statements about future research directions or potential clinical applications in the Discussion section (page 21 lines 81-88).

Reviewer #2:

Comment 1�The abstract could provide more specific details about the magnitude of improvements observed, particularly in the combined treatment group, to give readers a clearer sense of the study's impact.

Response 1� Thank you for your constructive comment. As suggested, we have added specific details on the extent of improvement in each indicator, particularly in the combination treatment group (page 2 lines 30-33).

Comment 2�The keywords are relevant. Introduction: The section could also benefit from a clearer statement of the study's novelty or how it advances existing knowledge in the field.

Response 2�Thank you for your constructive comments. As suggested, we added the novelty and how it advances existing knowledge in the field (pages 5 lines 95-111).

Comment 3�The methods section is detailed but could be improved in several areas:

The ethical statement is clear but could be more specific about the measures taken to minimize animal suffering, such as details on anesthesia or euthanasia protocols.

Response 3�As suggested, we added the details of the euthanasia protocol (page 6 lines 24-26).

Comment 4: The RNA-seq analysis lacks sufficient detail, particularly regarding the bioinformatics pipeline and the criteria used for identifying differentially expressed genes (DEGs).

Response 4: Thank you for your positive and constructive comment. As suggested, the description of the bioinformatics pipeline and the criteria for RNA-seq analysis has been refined and updated in detail (page 10 lines 05-17).

Comment 5�The results are presented clearly but could benefit from a more detailed discussion of the statistical significance, particularly in the context of the RNA-seq data. Some figures, while well-designed, have legends that could be more informative, particularly in explaining the significance of the findings.

Response 5�Thank you for this constructive comment. As suggested, we provided a more comprehensive discussion of the information presented in the relevant figures in the context of the RNA-seq data, and added an explanation of their significance (page 14 lines 16-24, page 15 lines 38-42, page 16 lines 50-62 and pages 16-17 lines 75-87).

Comment 6�The authors could provide a more critical evaluation of the study's limitations, such as the generalizability of the findings to humans or the potential for off-target effects of the interventions in the discussion.

Response 6�Thank you for your constructive comments. As suggested, we have added to the discussion section an assessment of the limitations of the study (page 21 lines 81-88).

Comment 7�The references are generally appropriate but could be more up-to-date, particularly in the discussion of the mechanisms of sarcopenia and the role of nutritional interventions. There are also inconsistencies in formatting, such as the use of italics and capitalization, which should be standardized.

Response 7�The references have been updated and standardized as suggested.

---

## [Decision Letter · Decision Letter 1]

12 May 2025

Lactoferrin combined with Coenzyme Q10 ameliorate sarcopenia in an aging mouse model induced by D-galactose

PONE-D-24-60346R1

Dear Dr. Luo,

We’re pleased to inform you that your manuscript has been judged scientifically suitable for publication and will be formally accepted for publication once it meets all outstanding technical requirements.

Kind regards,

Ayman A Swelum

Academic Editor

PLOS ONE

Additional Editor Comments (optional):

Reviewers' comments:

Reviewer's Responses to Questions

**Comments to the Author**

Reviewer #2: All comments have been addressed

Reviewer #3: All comments have been addressed

2. Is the manuscript technically sound, and do the data support the conclusions?

Reviewer #2: Yes

Reviewer #3: Yes

3. Has the statistical analysis been performed appropriately and rigorously?

Reviewer #2: I Don't Know

Reviewer #3: I Don't Know

4. Have the authors made all data underlying the findings in their manuscript fully available?

Reviewer #2: Yes

Reviewer #3: Yes

5. Is the manuscript presented in an intelligible fashion and written in standard English?

Reviewer #2: Yes

Reviewer #3: Yes

Reviewer #2: I appreciate the author's responsiveness and commitment to enhancing the quality of the work based on the feedback provided.

Reviewer #3: The respected authors addressed all comments and revised the manuscript carefully. The manuscript is presented well.

**Do you want your identity to be public for this peer review?** For information about this choice, including consent withdrawal, please see our Privacy Policy

Reviewer #2: No

Reviewer #3: No

---

## [Editor Report · Acceptance letter]

PONE-D-24-60346R1

PLOS ONE

Dear Dr. Luo,

I'm pleased to inform you that your manuscript has been deemed suitable for publication in PLOS ONE. Congratulations! Your manuscript is now being handed over to our production team.

Kind regards,

on behalf of

Professor Ayman A Swelum

Academic Editor

PLOS ONE